# Sea Slug Mucus Production Is Supported by Photosynthesis of Stolen Chloroplasts

**DOI:** 10.3390/biology11081207

**Published:** 2022-08-12

**Authors:** Diana Lopes, Sónia Cruz, Patrícia Martins, Sónia Ferreira, Cláudia Nunes, Pedro Domingues, Paulo Cartaxana

**Affiliations:** 1ECOMARE—Laboratory for Innovation and Sustainability of Marine Biological Resources, CESAM—Centre for Environmental and Marine Studies, Department of Chemistry, University of Aveiro, 3810-193 Aveiro, Portugal; 2ECOMARE, CESAM, Department of Biology, University of Aveiro, 3810-193 Aveiro, Portugal; 3CICECO—Aveiro Institute of Materials, Department of Chemistry, University of Aveiro, 3810-193 Aveiro, Portugal; 4CICECO, Department of Materials and Ceramic Engineering, University of Aveiro, 3810-193 Aveiro, Portugal; 5Mass Spectrometry Centre, LAQV-REQUIMTE—Associated Laboratory for Green Chemistry of the Network of Chemistry and Technology, Department of Chemistry, University of Aveiro, 3810-193 Aveiro, Portugal

**Keywords:** carbohydrates, *Elysia crispata*, kleptoplasty, Sacoglossa

## Abstract

**Simple Summary:**

Kleptoplasty is the capacity of a non-photosynthetic organism to acquire and maintain structurally intact chloroplasts from its algal food source(s), thereafter termed kleptoplasts. In animals, the capacity for long-term (several weeks to months) maintenance of photosynthetic active kleptoplasts is a unique characteristic in a handful of sea slugs, mostly within the genus *Elysia*. In this study, we investigated the role of kleptoplast photosynthesis on mucus production by the tropical sea slug *Elysia crispata*. Mucus secretion is paramount for mollusks, playing a role in locomotion, feeding, reproduction and protection. Limiting photosynthesis by rearing animals under reduced light led to lower mucus production and lower carbohydrate concentrations in the secreted mucus. This study indicates that production of mucus by kleptoplast-bearing sea slugs is supported by photosynthesis, confirming the biological relevance of kleptoplasty to the fitness of these peculiar mollusks.

**Abstract:**

A handful of sea slugs of the order Sacoglossa are able to steal chloroplasts—kleptoplasts—from their algal food sources and maintain them functionally for periods ranging from several weeks to a few months. In this study, we investigated the role of kleptoplast photosynthesis on mucus production by the tropical sea slug *Elysia crispata*. Animals reared for 5 weeks in quasi dark (5 μmol photons m^−2^ s^−1^) showed similar growth to those under regular light (60–90 μmol photons m^−2^ s^−1^), showing that kleptoplast photosynthesis was not relevant for growth when sea slugs were fed ad libitum. However, when subjected to short-term desiccation stress, animals reared under regular light produced significantly more mucus. Furthermore, the carbohydrate content of secreted mucus was significantly lower in slugs limited in the photosynthetic activity of their kleptoplasts by quasi-dark conditions. This study indicates that photosynthesis supports the synthesis of protective mucus in kleptoplast-bearing sea slugs.

## 1. Introduction

Sacoglossan sea slugs are a group of herbivorous marine gastropods. A striking feature of some of these sea slugs is their ability to digest the cellular components in algal food while retaining functional chloroplasts [1]. The stolen chloroplasts—kleptoplasts—continue to photosynthesize within the cells of the digestive tubules, providing sea slugs with additional resources to withstand food shortages or to invest in reproduction [2,3]. Within the metazoans, the ability to maintain long-term (several weeks to months) structurally intact and photosynthetic competent chloroplasts is a unique feature in a handful of Sacoglossa [4].

*Elysia crispata* Mörch, 1863, is one of the largest species of Sacoglossa in the Caribbean and the Gulf of Mexico, capable of feeding on different macroalgae, from which it sequesters long-term functional chloroplasts [5,6]. Distinct from reef-inhabiting *E. crispata*, the mangrove-dwelling slugs of the Florida Keys were described as *E. clarki* [7]. However, subsequent molecular analysis synonymized *E. clarki* with *E. crispata* [8]. This sea slug species can reach up to 15 cm in length, producing copious amounts of mucus under stressful conditions [9]. Water, inorganic salts and protein–polysaccharide complexes are the main components in the mucus of marine mollusks, the latter conferring the mucus its main properties [10]. The mucus may play a role in several physiological processes, such as locomotion, feeding, reproduction and protection, including reducing exposure to physical stress and predation [10].

Previous studies have shown the incorporation of carbon derived from kleptoplast photosynthesis into the secreted mucus of sacoglossan sea slugs [11,12]. In the present study, we investigated a putative role of kleptoplast photosynthesis in the production of mucus by the sea slug *E. crispata* under desiccation stress, determining the effects of limiting photosynthesis on the amount and total carbohydrate composition of the secreted mucus. We report experimental evidence supporting a relevant role of photosynthesis in the production of protective mucus by kleptoplast-bearing sea slugs.

## 2. Materials and Methods

### 2.1. Sea Slug Maintenance and Experimental Setup

Adult specimens of *E. crispata*, reported to be collected in Florida, USA, were purchased from TMC Iberia (Lisbon, Portugal). Sea slugs were maintained in 150 L recirculated life-support systems (LSS) operated with artificial seawater (ASW) at 25 °C and a salinity of 35. The photoperiod was set to 12 h light:12 h dark, with a photon scalar irradiance of 60 μmol photons m^−2^ s^−1^ being provided by T5 fluorescent lamps. Irradiance was measured with a Spherical Micro Quantum Sensor and a ULM-500 Universal Light Meter (Heinz Walz GmbH, Effeltrich, Germany). The animals were fed with the macroalga *Bryopsis plumosa*. The alga was grown in 2 L flasks with f/2 medium (without silica) and constant aeration at an irradiance of 100 μmol photons m^−2^ s^−1^. Spawned egg masses were collected and transferred to individual 250 mL Erlenmeyers with autoclaved ASW and gentle aeration, maintained in a growth chamber under the same photoperiod, temperature, salinity and light conditions as described above. Recently hatched juveniles were transferred to new Erlenmeyers and offered *B. plumosa*. The animals started feeding and acquiring chloroplasts. In the early stages of growth, the juveniles were first moved into 2 L plastic containers and when reaching 20–30 mm in length, the sea slugs were transferred to the LSS.

Thirty, two-month-old, first-generation laboratory-reared sea slugs were placed in individual wells within an LSS (see [13] for details) and maintained for five weeks under the same conditions as described above, but under two different irradiances during the light period. Fifteen animals were placed under regular light (60–90 μmol photons m^−2^ s^−1^) and the other fifteen under quasi dark (5 μmol photons m^−2^ s^−1^). Extremely low light intensity, sufficient to severely limit kleptoplast photosynthesis, was used instead of complete darkness so as not to disturb daily biorhythms and animal behavior (see [3] for details). Animals were fed ad libitum by providing each sea slug with approximately 200 mg fresh weight of *B. plumosa* every other day.

### 2.2. Sea Slug Fresh Weight

The sea slugs were weighed at the start of the experiment and then weekly during the 5-week experimental period. To determine the fresh weight, the animals were transferred to a 200 µm pore-sized sieve, which was placed on top of tissue paper to remove excess of water. The animals were weighed and returned to their individual wells in the LSS.

### 2.3. Mucus Collection and Carbohydrate Analysis

Mucus collection was performed at the end of the 5-week experimental period. After fresh weight assessment, animals were wetted with 200 µL ASW and placed in individual watch glasses for 5 min in dim light and at room temperature (24 °C). Mucus production was stimulated by gently swirling the animal in the watch glass. The mucus produced by each animal was collected using a micropipette, transferred to Eppendorf tubes, weighed, frozen in liquid nitrogen and stored at –80 °C. Total carbohydrate analysis was carried out on thawed samples according to the Dubois colorimetric procedure [14] using glucose as standard. Absorbances were recorded at 490 nm using an Eon microplate reader (Biotek, Winooski, VT, USA).

### 2.4. Statistical Analysis

Significant differences in sea slug fresh weight were tested using a mixed ANOVA with time as a within-subjects factor and light treatment as a between-subjects factor. Statistically significant differences in the amount and total carbohydrate composition of mucus produced by sea slugs from the two different light treatments were tested using non-parametric Mann–Whitney U tests. A Pearson correlation was run to determine the relationship between the amount of mucus produced and sea slug weight. Statistical analysis was carried out using IBM SPSS Statistics 28.

## 3. Results

*Elysia crispata* fed ad libitum over the 5-week experimental period exhibited similar growth patterns under regular light and quasi-dark conditions (Figure 1). A significant increase in sea slug fresh weight was observed over time (F_5,140_ = 82.402, *p* < 0.001), with an average increase of 54%. There was no significant effect of light treatment on sea slug fresh weight (F_1,28_ = 0.178, *p* = 0.676).

Under short-term desiccation stress, *E. crispata* reared under regular light produced significantly more mucus (U = 34.0, *p* = 0.001) than conspecifics raised in quasi dark (Figure 2a). When assessing the carbohydrate content, significantly higher concentrations (U = 17.5, *p* < 0.001) were observed in the secreted mucus of sea slugs reared under regular light than in quasi dark (Figure 2b). When sampling the mucus, it was clear that the secretions from sea slugs reared under regular light were more viscous than those from quasi-dark-raised animals. Under normal light conditions, the sea slugs produced 49% more mucus, which was 115% richer in carbohydrates.

There was no significant correlation between the amount of mucus produced and sea slug weight (r = 0.217, *n* = 30, *p* = 0.249).

## 4. Discussion

The growth of *E. crispata* was similar under the different light conditions tested (regular light vs. quasi-dark), showing that kleptoplast photosynthesis was not relevant for growth when sea slugs had unrestricted access to their macroalgal food source. Using stable nitrogen isotopic composition of amino acids, Maeda et al. [15] reported that kleptoplast photosynthesis represented a significant nutritional source for sea slugs during starvation, but the contribution was negligible for specimens from natural habitats where food algae are plentiful.

Here, under desiccation stress, sea slugs reared under regular light showed significantly higher mucus production and carbohydrate concentrations in the secreted mucus than animals raised under quasi dark. Observed differences in mucus production were not attributable to sea slug size. Our results suggest that kleptoplast photosynthesis provides substrates for mucus synthesis by the animal, affecting not only the amount of mucus produced but also its properties (e.g., viscosity).

Light-dependent incorporation of inorganic carbon and translocation to various kleptoplast-free tissues and the secreted mucus have been reported in Sacoglossa sea slugs [3,11,12,16,17]. Using high-resolution secondary ion mass spectrometry (NanoSIMS), labelled ^13^C fixed by kleptoplast photosynthesis in the digestive tubules was imaged within 6 h in reproductive organs (albumen gland and gonadal follicles) of the sea slugs *E. timida* and *E. viridis* [3,16]. Using labelled ^14^C, Trench et al. [17] showed carbon incorporation in several tissues of *E. crispata* and *E. diomedea*, including the mucus-secreting pedal gland. About 5 to 10% of the carbon fixed by kleptoplast photosynthesis in the sea slug *Plakobranchus ocellatus* was later detected in the secreted mucus [11]. Trench et al. [12] estimated that about 30% of the total labelled carbon fixed by the kleptoplasts of *E. crispata* and *E. diomedea* was recovered in the mucus. After acid hydrolysis of the mucus, most of the labelled carbon was incorporated into galactose and glucose, likely part of sulfated polysaccharides [11,12].

Accordingly, the release of mucus from symbiotic reef corals has often been associated with phototrophic nutrition [18,19]. Higher light levels lead to increased photosynthetic activity in zooxanthellae and greater translocation to the coral host of high-energy compounds. Under these conditions, the energy-rich (but nitrogen-poor) algal photosynthates are channeled into coral respiration and mucus production [19,20]. In a shallow-water coral, Davies [20] estimated that about 48% of the photosynthetically fixed carbon by the zooxanthellae on a bright day was excreted as mucus. Mucus polysaccharides can screen the coral and their zooxanthellae from the deleterious physiological impacts of excessive solar irradiance, maintaining the stability and functioning of the intact symbiosis [21]. In sacoglossan sea slugs, UV screening and singlet oxygen protective polypropionates, synthesized with carbon obtained from photosynthesis, may contribute to the maintenance of long-term photosynthetic activity in kleptoplasts [22,23].

## 5. Conclusions

The light-dependent incorporation of inorganic carbon into the mucus reported in previous studies shows that sea slugs use kleptoplast-derived carbohydrates for mucus production. In this study, the observation that *E. crispata* reared under regular light secreted over three-times as many carbohydrates as those raised under quasi dark suggests that carbon supply from kleptoplast photosynthesis is quantitatively important for mucus synthesis. Due to the importance of mucus secretion for several physiological processes in mollusks, our results underline the biological relevance of photosynthesis to the overall fitness of kleptoplast-bearing sea slugs. Further studies are required to determine the relevance of the secreted mucus on the long-term maintenance of functional kleptoplasts.

## Figures and Tables

**Figure 1 biology-11-01207-f001:**
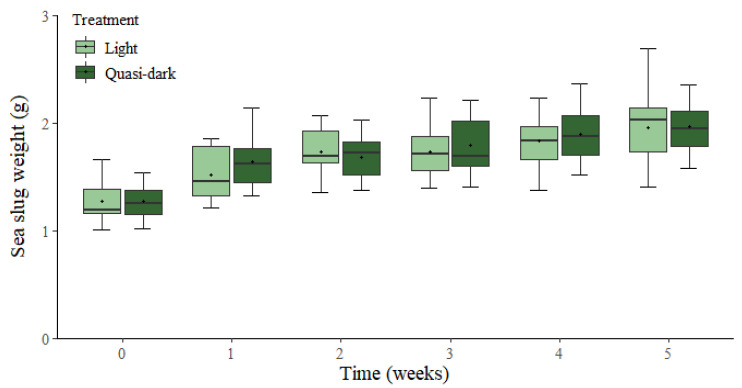
Weight change in the sea slug *Elysia crispata* reared for five weeks under light (60–90 µmol photons m^−2^ s^−1^) and quasi dark (5 µmol photons m^−2^ s^−1^). The sea slugs were fed ad libitum with the macroalga *Bryopsis plumosa*. The line is the median, the **^+^** represents the mean, top and bottom of the box are the 75% and 25% percentiles, respectively, and the whiskers represent the maximum and minimum values (*n* = 15).

**Figure 2 biology-11-01207-f002:**
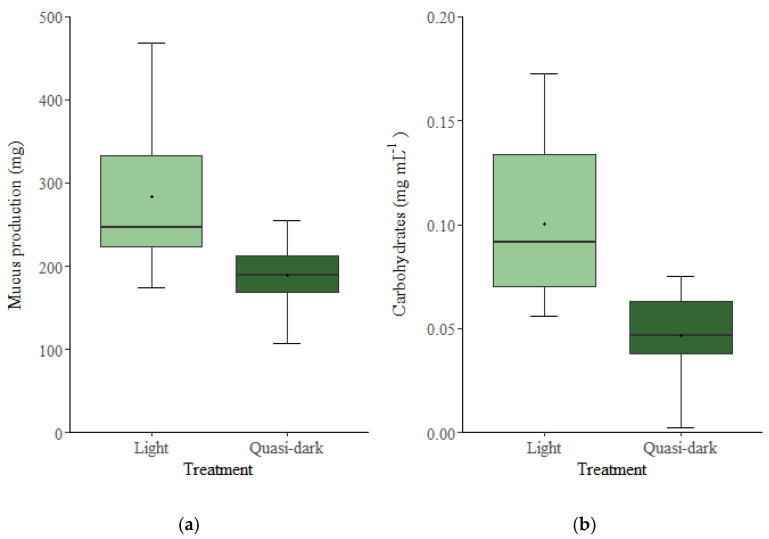
Mucus production (**a**) and mucus total carbohydrate content (**b**) in the sea slug *Elysia crispata* reared for five weeks under light (60–90 µmol photons m^−2^ s^−1^) and quasi dark (5 µmol photons m^−2^ s^−1^). The sea slugs were fed ad libitum with the macroalga *Bryopsis plumosa*. The line is the median, the **^+^** represents the mean, top and bottom of the box are the 75% and 25% percentiles, respectively, and the whiskers represent the maximum and minimum values (*n* = 15).

## Data Availability

The data are provided in the electronic Appendix A.

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
