# Peer review of "Sea Slug Mucus Production Is Supported by Photosynthesis of Stolen Chloroplasts"

_biology, 2022, doi:10.3390/biology11081207_

Round 1

Reviewer 1 Report

Kleptoplasty by sea slugs (animal photosynthesis) draws tremendous interest from the broader scientific community and public, and has been a highly controversial subject for some years now. Numerous high-profile publications have recently asserted, with little evidence, that kleptoplasty harms or kills the host slug and reflects some loss of digestional ability; later revisions to these claims assert that stolen chloroplasts may not be harmful but only allow animals to survive starvation. Here, the authors contribute a nice addition to the growing body of experimental evidence showing that, contrary to these claims, kleptoplasty is an important part of the natural biology of these slugs and that slugs are adapted to benefit from photosynthetically fixed carbon. Here, the experimental evidence shows creatively that without the benefit of light, even allowed to feed ad libitum, slugs cannot maintain their normal mucus production. Although prior work showed that fixed carbon gets translocated into pedal mucus in these animals, I have not seen experiments performed that manipulated light in this context. I thought it was an interesting result worthy of publication, and I offer some comments to improve the paper.

To a mollusc, mucus is life: these slimy animals rely on mucus for crawling and gliding on their muscular foot, for surviving desiccation stress, to secrete deterrent compounds and resist predation, to encase egg masses, and for myriad other purposes. I recommend the authors expand more generally on why mucus is so important to snails and slugs so the general reader understands this. I also would strengthen the connection in the discussion between kleptoplasty and the role of mucus in slug survival. To me, the take-home message of these results is that slugs are adapted to exploit the fixation of carbon by their plastids and do not invest their own energy in the biosynthesis of mucus carbohydrates (or cannot), so in the dark they become highly depleted in mucus reserves; this in turn shows how dependent on their photosynthetic ‘partners’ the slug host really is, contrary to claims that slugs suffer in this holobiont. I’m not sure this message comes through clearly in the discussion.

Introduction, lines 43-50: most of the sentences in this opening paragraph are not exactly correct. For instance, not all sacoglossans eat macroalgae; some eat diatoms, or seagrass (angiosperms), or mollusc eggs. Only about half of sacoglossans retain functional chloroplasts (the second sentence indicates all sacoglossans do this). Kleptoplasty is no longer unique to sacoglossans among animals, as some flatworms are now known to keep red algal plastids, although it isn’t clear for how. The introduction needs to be carefully reworded for accuracy and precision.

Lines 50-51: There is no basis for saying E. crispata is one of the most abundant Caribbean sacoglossans; this is almost certainly not true. It is the most conspicuous because of its very large size, but smaller species are far more numerically abundant. Please clarify this, and cite studies to support however you want to frame this.

Author Response

Response to Reviewers

Reviewer #1:

Kleptoplasty by sea slugs (animal photosynthesis) draws tremendous interest from the broader scientific community and public, and has been a highly controversial subject for some years now. Numerous high-profile publications have recently asserted, with little evidence, that kleptoplasty harms or kills the host slug and reflects some loss of digestional ability; later revisions to these claims assert that stolen chloroplasts may not be harmful but only allow animals to survive starvation. Here, the authors contribute a nice addition to the growing body of experimental evidence showing that, contrary to these claims, kleptoplasty is an important part of the natural biology of these slugs and that slugs are adapted to benefit from photosynthetically fixed carbon. Here, the experimental evidence shows creatively that without the benefit of light, even allowed to feed ad libitum, slugs cannot maintain their normal mucus production. Although prior work showed that fixed carbon gets translocated into pedal mucus in these animals, I have not seen experiments performed that manipulated light in this context. I thought it was an interesting result worthy of publication, and I offer some comments to improve the paper.

To a mollusc, mucus is life: these slimy animals rely on mucus for crawling and gliding on their muscular foot, for surviving desiccation stress, to secrete deterrent compounds and resist predation, to encase egg masses, and for myriad other purposes. I recommend the authors expand more generally on why mucus is so important to snails and slugs so the general reader understands this. I also would strengthen the connection in the discussion between kleptoplasty and the role of mucus in slug survival. To me, the take-home message of these results is that slugs are adapted to exploit the fixation of carbon by their plastids and do not invest their own energy in the biosynthesis of mucus carbohydrates (or cannot), so in the dark they become highly depleted in mucus reserves; this in turn shows how dependent on their photosynthetic ‘partners’ the slug host really is, contrary to claims that slugs suffer in this holobiont. I’m not sure this message comes through clearly in the discussion.

Reply: Changes were made to the Simple Summary and Conclusions to stress the relevance of mucus production to the slugs and of the animal-organelle partnership, as suggested by the reviewer.

Introduction, lines 43-50: most of the sentences in this opening paragraph are not exactly correct. For instance, not all sacoglossans eat macroalgae; some eat diatoms, or seagrass (angiosperms), or mollusc eggs. Only about half of sacoglossans retain functional chloroplasts (the second sentence indicates all sacoglossans do this). Kleptoplasty is no longer unique to sacoglossans among animals, as some flatworms are now known to keep red algal plastids, although it isn’t clear for how. The introduction needs to be carefully reworded for accuracy and precision.

Reply: The first sentence of the Introduction was changed according to the reviewer’s indications. Regarding the second sentence, it is stated that “A striking feature of SOME of these sea slugs is their ability…”, so the reviewer perception that it refers to all sacoglossans is not correct. What is said in the last sentence of the first paragraph of the Introduction, “that long-term (several weeks to months) structurally intact and photosynthetic competent chloroplasts is a unique feature of a handful of Sacoglossa”, is exact. What has been shown for rhabdocoel flatworms is short-term (only a few days) maintenance of active chloroplasts. 

Lines 50-51: There is no basis for saying E. crispata is one of the most abundant Caribbean sacoglossans; this is almost certainly not true. It is the most conspicuous because of its very large size, but smaller species are far more numerically abundant. Please clarify this, and cite studies to support however you want to frame this.

Reply: The sentence was changed, as suggested by the reviewer.

Reviewer 2 Report

Lopes et al. reported that light conditions affect the mucus production of kleptoplastic sea slug Elysia crispata. Their simple and compelling experiments reveal a new aspect of the reaction of this species against light illumination. In my opinion, however, there are several leaps in the discussion which should be corrected before publication in the "Biology" journal. Especially, it should be clarified that the source of the increased mucus keeps in undetermined in their analysis. 

The title, "supported by photosynthesis", is improper, and the authors have not provided adequate evidence for their major claims that "sea slugs use kleptoplast-derived metabolites for mucus production". Previous studies have shown that kleptoplastic photosynthetic products are used as mucus in several sea slug species, as described in the MS. If the authors claim that observed mucus increment is dependent on photosynthesis, they should at least trace photosynthetic products using isotopes chasing as in previous works. It is true that their results showed that prior (enough) light exposure altered the sea slug's response to physical stress. However, this result can explain that E. crispata changed its defense strategy depending on light conditions. This MS gave no discussion about this possibility. The latter hypothesis seems more promising if photosynthesis has no trophic contribution.

The authors discussed the fewness of the nutritional effect of photosynthesis based on the wet weight. However, the wet weight is not a reliable index of the trophic contribution of photosynthesis. The authors should specify that accurate nutritional assessment requires for reliable discussion, including isotopic analysis and spawning observation. 

In addition, several previous studies have already shown the limited nutritional effect of kleptoplasty under feeding conditions, and these should be cited and discussed with the authors result (https://www.nature.com/articles/s41598-017-08002-0 , https://link.springer.com/article/10.1007/s00227-014-2402-1, https://journals.plos.org/plosone/article?id=10.1371/journal.pone.0042024). The authors' data is important and attractive enough. I am unaware of any studies focusing on mucus production under experimental feeding conditions. I believe this study's significance will be more apparent if the author describes the relationship with the previous studies clearly. 

E. crispata is one of the "long-term" species. However, their retention period is shorter than other "long-term" species (e.g., Elysia chlorotica, Plakobranchus ocellatus). Previous research about those long "long term" species gave several discussions about the nutritional contribution. 

The authors' results do not evaluate the general feature of kleptoplasty phenomena but the E. chrispata-specific reaction. Therefore, it would be more appropriate to change the description of the "kleptoplasty" to "kleptoplasty in E. crispata". 

e.g. Line 146

"that kleptoplast photosynthesis was not relevant for growth when sea slugs had unrestricted access to their macroalgal food source."

The data plotting via box plot should be changed to the dot or jitter plot (suitable and modern format to reflect the actual data distribution.

The authors should discuss the relationship between the sample body mass and extracted mucus mass. If Tables S1 and S2 use the same sample set, the correlation between body weight and mucus production would be important data.

Author Response

Reviewer #2

Lopes et al. reported that light conditions affect the mucus production of kleptoplastic sea slug Elysia crispata. Their simple and compelling experiments reveal a new aspect of the reaction of this species against light illumination. In my opinion, however, there are several leaps in the discussion which should be corrected before publication in the "Biology" journal. Especially, it should be clarified that the source of the increased mucus keeps in undetermined in their analysis.

The title, "supported by photosynthesis", is improper, and the authors have not provided adequate evidence for their major claims that "sea slugs use kleptoplast-derived metabolites for mucus production". Previous studies have shown that kleptoplastic photosynthetic products are used as mucus in several sea slug species, as described in the MS. If the authors claim that observed mucus increment is dependent on photosynthesis, they should at least trace photosynthetic products using isotopes chasing as in previous works. It is true that their results showed that prior (enough) light exposure altered the sea slug's response to physical stress. However, this result can explain that E. crispata changed its defense strategy depending on light conditions. This MS gave no discussion about this possibility. The latter hypothesis seems more promising if photosynthesis has no trophic contribution.

Reply: The reviewer refers that we do not show incorporation of photosynthetic products in the mucus. Although this is correct, incorporation of photosynthetic-derived compounds has been shown before for several photosynthetic sea slugs, including Elysia crispata (Trench et al. 1970; 1972). Furthermore, isotope tracking revealed incorporation in carbohydrates (Trench et al. 1970; 1972), which were quantified in our study. In order to clarify this issue for the reader, we have now included reference to these two studies in the last paragraph of the Introduction, whereas in the previous version they appeared for the first time in the Discussion. In the first sentence of the Conclusions, we added “shown in previous studies” to clear identify “that sea slugs use kleptoplast-derived metabolites for mucus production” is not a direct conclusion of our work.

The authors discussed the fewness of the nutritional effect of photosynthesis based on the wet weight. However, the wet weight is not a reliable index of the trophic contribution of photosynthesis. The authors should specify that accurate nutritional assessment requires for reliable discussion, including isotopic analysis and spawning observation.

In addition, several previous studies have already shown the limited nutritional effect of kleptoplasty under feeding conditions, and these should be cited and discussed with the authors result (https://www.nature.com/articles/s41598-017-08002-0 , https://link.springer.com/article/10.1007/s00227-014-2402-1, https://journals.plos.org/plosone/article?id=10.1371/journal.pone.0042024). The authors' data is important and attractive enough. I am unaware of any studies focusing on mucus production under experimental feeding conditions. I believe this study's significance will be more apparent if the author describes the relationship with the previous studies clearly.

Reply: We have included in the Discussion reference to the work of Maeda et al. 2012, as suggested by the reviewer. The work of Akimoto et al. 2014 refers to two short-term retention sacoglossans, so we did not include it. The work by Cartaxana et al. 2017 (our own work) was also not included because it does not compare light- and dark-reared fed specimens.

E. crispata is one of the "long-term" species. However, their retention period is shorter than other "long-term" species (e.g., Elysia chlorotica, Plakobranchus ocellatus). Previous research about those long "long term" species gave several discussions about the nutritional contribution.

The authors' results do not evaluate the general feature of kleptoplasty phenomena but the E. chrispata-specific reaction. Therefore, it would be more appropriate to change the description of the "kleptoplasty" to "kleptoplasty in E. crispata".

e.g. Line 146

"that kleptoplast photosynthesis was not relevant for growth when sea slugs had unrestricted access to their macroalgal food source."

Reply: In line 146, the term sea slugs was used not to repeat “E. crispata”, already at the beginning of the sentence. By reading the entire sentence, it is clear that we refer to E. crispata and not other kleptoplast-bearing sea slugs. We have changed “sea slugs” to “E. crispata” in the conclusions, as suggested by the Reviewer.

The data plotting via box plot should be changed to the dot or jitter plot (suitable and modern format to reflect the actual data distribution).

Reply: We disagree that dot or jitter plots are better representation of our data set, as we lose information of the average and median that box-whisker plots provide. Furthermore, box-whisker plots provide detailed information on data distribution, including 25 and 75% percentiles, as well as the upper and lower extreme values.

The authors should discuss the relationship between the sample body mass and extracted mucus mass. If Tables S1 and S2 use the same sample set, the correlation between body weight and mucus production would be important data.

Reply: Reference to the (absence of) correlation between body weight and mucus production was added to the Results, as suggested by the reviewer. Reference to the correlation test was included in the Statistical Analysis.